# $Co^2PT$: Mitigating Bias in Pre-trained Language Models through Counterfactual Contrastive Prompt Tuning

**Xiangjue Dong**[1]   **Ziwei Zhu**[2]   **Zhuoer Wang**[1]   **Maria Teleki**[1]   **James Caverlee**[1]

[1] Texas A&M University   [2] George Mason University

{xj.dong, wang, mariateleki, caverlee}@tamu.edu   zzhu20@gmu.edu

## Abstract

Pre-trained Language Models are widely used in many important real-world applications. However, recent studies show that these models can encode social biases from large pre-training corpora and even amplify biases in downstream applications. To address this challenge, we propose $Co^2PT$, an efficient and effective *debias-while-prompt tuning* method for mitigating biases via counterfactual contrastive prompt tuning on downstream tasks. Our experiments conducted on three extrinsic bias benchmarks demonstrate the effectiveness of $Co^2PT$ on bias mitigation during the prompt tuning process and its adaptability to existing upstream debiased language models. These findings indicate the strength of $Co^2PT$ and provide promising avenues for further enhancement in bias mitigation on downstream tasks.

## 1 Introduction

Pre-trained language models (PLMs) are widely used in many real-world applications, demonstrating remarkable performance (Devlin et al., 2019; Brown et al., 2020). However, it has been demonstrated that PLMs encode unfair social biases in their parameters based on their pre-training step over large-scale text corpora (May et al., 2019). Furthermore, these biases – for example, based on gender, race, or religion – can easily propagate to the downstream tasks that use these PLMs (Kaneko and Bollegala, 2021). For example, *"She is a nurse"* can have a higher conditional likelihood than *"He is a nurse"* in the language modeling task, and *"nurse"* can have higher coreference scores to *"she"* than *"he"* in the coreference resolution task (Lu et al., 2020). Considering that NLP applications like machine translation systems, resume filtering systems, dialogue systems, and speech recognition (Tatman, 2017) are widely used by millions of users globally, it is crucial to mitigate the social biases present in PLMs and strive for models that will not propagate discriminatory predictions or offensive outputs towards specific groups before being deployed.

Much prior effort has focused primarily on debiasing the representations learned during the pre-training process, e.g., through projection (Dev et al., 2020; Liang et al., 2020; Ravfogel et al., 2020; Kaneko and Bollegala, 2021), further pre-training on unbiased external corpora (Webster et al., 2020; Lauscher et al., 2021; He et al., 2022), or fine-tuning to debias (Cheng et al., 2021; Guo et al., 2022). The effectiveness of such debiasing efforts is typically measured on *intrinsic* benchmarks like SEAT (Sentence Encoding Association Test) which computes the association between demographic terms (e.g., woman, man) and stereotype terms (e.g., science, art). An unbiased model should display no difference in the similarity between the representations of these terms (May et al., 2019).

While these existing approaches help reduce social biases under intrinsic measures, these *debias-then-finetune* methods are based on the hypothesis that if an upstream model is unbiased, it will also preserve its fairness effects on downstream tasks during the fine-tuning process. However, recent research investigating the relationship between intrinsic and *extrinsic* benchmarks (which evaluate fairness in downstream applications) finds these two benchmarks correlate weakly (Kaneko et al., 2022). Furthermore, they observe that models, even after being debiased, tend to re-acquire or even amplify biases (e.g., instance-related biases and label-related biases) during the fine-tuning process on downstream tasks (Zhao et al., 2017; Leino et al., 2019). Thus, this mismatch leads to our motivating research question – *How can we develop an efficient and effective method to mitigate bias on downstream tasks?*

To answer the aforementioned question, we propose $Co^2PT$, a *debias-while-prompt tuning* approach through **Co**unterfactual **Co**ntrastive **P**rompt

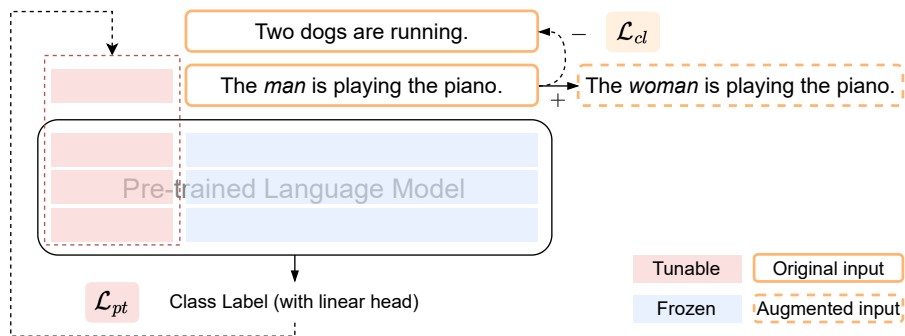

Figure 1: The overview of Co$^2$PT. First, we construct counterfactual pairs from the training data. Then, we learn debiased continuous prompts by simultaneously optimizing prompt tuning loss $\mathcal{L}_{pt}$ on downstream tasks and contrastive loss $\mathcal{L}_{cl}$ between the counterfactual pairs.

Tuning. In this method, we first freeze all parameters of the PLM and add tunable continuous prompts for every layer. Unlike the previous debias-then-finetune methods that require expensive re-training of the original PLM and risk knowledge forgetting, this deep prompt tuning framework saves computational and memory resources while preserving the original pre-trained knowledge and language modeling ability (Li and Liang, 2021; Liu et al., 2022). To ensure that a fair system generates unbiased results regardless of the demographic terms used, we construct counterfactual pairs directly from the training data, eliminating the need for external corpora that heavily depend on their quality for debiasing. Specifically, we replace demographic terms associated with either the dominant or minoritized group in the training data with terms representing the opposite group. Then, we integrate the ability to mitigate bias into the prompt parameters through a contrastive objective between counterfactual pairs while maintaining the parameters of PLMs frozen. Co$^2$PT can be integrated into existing debiased models to help them mitigate biases on downstream tasks and offer flexibility in addressing different kinds of bias. These advantages establish Co$^2$PT as an efficient and effective method for mitigating bias in downstream tasks.

In conclusion, the proposed Co$^2$PT mitigates bias on downstream tasks through prompt tuning, making the following contributions:

- Co$^2$PT achieves time and memory efficiency without requiring access to an external corpus or retraining the entire model.

- Over three extrinsic bias benchmarks, we show that Co$^2$PT effectively mitigates bias amplified during the prompt tuning process on downstream tasks.

- Furthermore, Co$^2$PT can be extended to existing debiased language models, effectively bridging the gap between debiased upstream models and downstream tasks.

## 2 Related Work

Several approaches have been proposed for debiasing pre-trained language models such as projection-based methods (Dev et al., 2020; Liang et al., 2020; Ravfogel et al., 2020; Kaneko and Bollegala, 2021), post-hoc text generation techniques (Schick et al., 2021), adversarial methods (Han et al., 2021), fine-tuning on biased prompts (Guo et al., 2022), with contrastive objective (Cheng et al., 2021) or with augmented data (Zhao et al., 2018), additional pre-training methods on re-balanced corpus through counterfactual data augmentation (Webster et al., 2020; Lauscher et al., 2021; Meade et al., 2022) or with a contrastive objective on gender-balanced entailment pairs (He et al., 2022), using dropout regularization (Webster et al., 2020), through parameter-efficient methods (Lauscher et al., 2021; Yang et al., 2022; Xie and Lukasiewicz, 2023) or with a contrastive objective (Li et al., 2023). While some works do not require access to an external corpus or do not require retraining the entire model, most prior methods primarily focus on mitigating bias within the model's intrinsic characteristics and evaluate the effectiveness of bias mitigation through intrinsic bias benchmarks, e.g., SEAT (May et al., 2019), StereoSet (Nadeem et al., 2021), and CrowS-Pairs (Nangia et al., 2020). Subsequently, they fine-tune the debiased models on downstream tasks and demonstrate that their debiased models retain the language modeling ability and the performance on downstream tasks or extrinsic bias benchmarks, which evaluate fairness in downstream tasks by

testing whether the models exhibit different performances among different populations.

Nevertheless, recent research shows that these *debias-then-finetune* methods will re-acquire or even amplify biases during the fine-tuning process on downstream tasks and that intrinsic and extrinsic evaluation bias benchmarks correlate poorly (Goldfarb-Tarrant et al., 2021; Cao et al., 2022; Kaneko et al., 2022). They encourage researchers to focus directly on extrinsic measures of bias of specific applications when addressing bias mitigation (Goldfarb-Tarrant et al., 2021).

Thus, we focus in this paper on mitigating bias on downstream tasks and evaluate using extrinsic evaluation benchmarks directly. In addition, different from the previous methods requiring further pre-training on the counterfactually augmented sentences from an external corpus, e.g., English Wikipedia (Zmigrod et al., 2019; Webster et al., 2020; Meade et al., 2022), BookCorpus (Lauscher et al., 2021), News-Commentary v15 (Yang et al., 2022) or NLI (He et al., 2022), our methods achieve time and memory efficiency by eliminating the need for external corpus access or model retraining.

## 3 Co$^2$PT: Debiasing via Counterfactual Contrastive Prompt Tuning

We propose Co$^2$PT, a *debias-while-prompt tuning* parameter-efficient method for mitigating biases on downstream tasks via counterfactual contrastive prompt tuning, presented in Figure 1. Concretely, Co$^2$PT mitigates bias in PLMs by leveraging counterfactual pairs from training data to produce debiased representations during prompt tuning.

**Deep Prompt Tuning.** First, we introduce the backbone framework of Co$^2$PT – deep prompt tuning. We incorporate continuous prompts as prefix tokens in every layer of the PLM. By doing this, we have more tunable task-specific parameters to enhance per-task capacity while maintaining parameter efficiency (Li and Liang, 2021; Liu et al., 2022; Wang et al., 2022; Dong et al., 2023a). Besides, it can achieve comparable performance to fine-tuning, outperforming methods that only add trainable continuous prompts into the input embedding layer (Lester et al., 2021; Liu et al., 2021), which underperform the fine-tuning methods, especially when the model size is not large (Liu et al., 2022). The prompt tuning loss of proposed Co$^2$PT on the downstream task is represented as $\mathcal{L}_{pt}$, e.g., cross-entropy loss for a classification task.

**Counterfactual Pairs Construction.** Then, the first key question is: *how to interject the debiasing capability into the continuous prompts?* An unbiased model should make the same predictions independent of the bias-attribute term, thus we apply counterfactual data augmentation to generate counterparts of training examples from the training data during prompt tuning. Concretely, let $S$ represent the training corpus and let $W = \{(w_1, w_2, \ldots, w_m)^i\}_{i=1}^N$ be a set of $N$ bias-attribute term pairs. For each sentence $s_i$ in $S$ and each pair $(w_1, w_2, \ldots, w_m)$ in $W$, for any $w_i$ in $s$, we replace it with the term along the opposite bias direction. Take the binary-gender debiasing task shown in Figure 1 for example, the bias-attribute terms are {(man, woman), (he, she), . . . }. The "man" is in the input sentence "The man is playing the piano". We replace it with "woman" while leaving non-attribute words unchanged. Then the counterfactually augmented sentence is "The woman is playing the piano", and vice versa. The obtained counterfactual sentence of the original sentence $s_i$ is denoted as $s'_i$.

**Counterfactual Contrastive Learning.** The counterfactual pairs construction allows us to achieve a balance in inputs containing bias-attribute terms. However, *how can we ensure that our model generates consistent predictions for both $s_i$ and $s'_i$, which possess similar semantic meaning but differ in bias direction?* To make the model generate predictions independent of biased attributes, it is important for sentences with similar semantics but along different bias directions to be closer (Cheng et al., 2021; He et al., 2022). We apply contrastive learning, of which the objective is to obtain meaningful representations by bringing semantically similar neighbors closer and pushing apart the dissimilar neighbors (Gao et al., 2021; Dong et al., 2023b; Li et al., 2023). In this work, input sentence $s_i$ and its counterpart $s'_i$ are semantically related but in opposite bias directions. We let $\mathbf{h}_i$ and $\mathbf{h}'_i$ denote the representations of $s_i$ and $s'_i$ and then concatenate with the continuous prompt representation $\mathbf{p}$ as positive pairs. Then we take the cross-entropy objective with in-batch negatives (Gao et al., 2021). The training objective for $(\mathbf{h}_i, \mathbf{h}'_i)$ with a mini-batch of $N$ pairs is:

$$\mathcal{L}_{cl} = -\log \frac{e^{\text{sim}(\mathbf{p} \oplus \mathbf{h}_i, \mathbf{p} \oplus \mathbf{h}'_i)/\tau}}{\sum_{j=1}^N e^{\text{sim}(\mathbf{p} \oplus \mathbf{h}_i, \mathbf{p} \oplus \mathbf{h}'_j)/\tau}}, \quad (1)$$

where $\text{sim}(\mathbf{x}_i, \mathbf{y}_i)$ is the cosine similarity of $\mathbf{x}_i$ and

$\mathbf{y}_i$: $\text{sim}(\mathbf{x}_i, \mathbf{y}_i) = \mathbf{x}_i^\top \mathbf{y}_i / \|\mathbf{x}_i\| \|\mathbf{y}_i\|$, $\oplus$ is the concatenation of two representations, and $\tau$ is a temperature hyperparameter.

For counterfactual pairs $(s_i, s_i')$ in the single-sentence classification task, $s_i$ is the original sentence from the training data and $s_i'$ is the augmented sentence that has the same semantic meaning as $s_i$ but in a different bias direction. For sentence-pair classification, like in the SNLI task, with $x_i$ as the premise and $y_i$ as the hypothesis, $s_i$ is the original premise-hypothesis pair $(x_i, y_i)$ while $s_i'$ is the counterfactual augmented premise-hypothesis pair $(x_i', y_i')$. Similarly, the sentence representations are concatenated with continuous prompts to calculate the contrastive loss through Equation 1.

**Learning Objectives.** Finally, the continuous prompts *learn-to-debias* by simultaneously optimizing the prompt tuning loss $\mathcal{L}_{pt}$ on downstream tasks and contrastive loss $\mathcal{L}_{cl}$ between the counterfactual pairs:

$$\mathcal{L} = \mathcal{L}_{pt} + \alpha \mathcal{L}_{cl}, \tag{2}$$

where $\alpha$ is a tunable coefficient hyperparameter. As stated before, we only tune the parameters of the debiasing continuous prompts while maintaining the parameters of PLMs frozen throughout the training. After the counterfactual contrastive prompt tuning, the debiasing knowledge is stored in the prompt parameters. This approach not only retains the knowledge within the original parameters of PLMs but is also flexible and adaptable to different downstream tasks. For example, we can train different prompts for different bias dimensions such as gender, race, and religion. These prompts can then be combined and applied to downstream tasks. Moreover, considering that prior research primarily concentrates on binary gender, it is more efficient to extend its application to non-binary gender without requiring re-training new debiased models.

## 4 Experimental Setup

We design experiments to test the effectiveness of our proposed $\text{Co}^2\text{PT}$ approach toward answering four questions: **RQ1**: Will $\text{Co}^2\text{PT}$ mitigate bias on downstream tasks effectively? **RQ2**: How will the existing intrinsic debiased methods perform on the downstream tasks when they are combined with $\text{Co}^2\text{PT}$? **RQ3**: What impact do different modules have on the design of $\text{Co}^2\text{PT}$? **RQ4**: How do hyperparameters affect $\text{Co}^2\text{PT}$?

### 4.1 Bias Evaluation

Extrinsic bias benchmarks assess bias via performance gap between different groups in downstream tasks. In this work, we evaluate $\text{Co}^2\text{PT}$ on three widely used extrinsic bias benchmarks: Bias-STS-B, Bias-NLI, and Bias-in-Bios.

**Bias-STS-B** (Webster et al., 2020) is adapted from the STS-B task to evaluate gendered correlations, which requires models to predict the semantic similarity between pairs of sentences. Specifically, 276 sentences are collected from the test set as templates and then gendered terms *(man, woman)* and professional terms from Rudinger et al. (2018) are inserted into each template, forming 16,980 sentence pairs. For instance, if the template is "A man is walking", then the sentence pairs are ("A man is walking", "A nurse is walking") and ("A woman is walking", "A nurse is walking"). If a model is unbiased towards gender terms, it should assign equal similarity scores to both pairs. We calculate the *average absolute difference* between the similarity scores of sentence pairs containing male and female terms, and how often the difference between "male" and "female" sentence pairs $> \tau$, where we report the results for $\tau = 0.1$ and $\tau = 0.3$ (Webster et al., 2020). A lower value indicates less bias.

**Bias-NLI** (Dev et al., 2020) is a natural language inference dataset consisting of neutral sentence pairs to evaluate the gender-occupation bias. It is constructed by populating the template: "The subject verb a/an object", leading to 1,936,512 instances. Concretely, the verb and object slots are filled with activities, e.g., *ate a bagel*. Then they create neutral entailment pairs by filling the subject slot with an occupation term with a strong gender correlation, e.g., "nurse", for the hypothesis, and "woman", for the premise, resulting in the instance: *The woman ate a bagel; The nurse ate a bagel. neutral.* Bias is defined as deviation from neutrality and measured by three metrics: (1) Net Neutral (NN): the average probability that the model assigns a neutral label for all instances, (2) Fraction Neutral (FN): the percentage of instances that the model predicts the neutral label and (3) Threshold: $\tau$ (T: $\tau$): the fraction of examples whose probability of neutral are above $\tau$. We report the results for $\tau = 0.5$ and $\tau = 0.7$ following (Lauscher et al., 2021; He et al., 2022). All three metrics will attain 1 for a bias-free model.

**Bias-in-Bios** (De-Arteaga et al., 2019) is a large-scale English dataset studying gender bias in oc-

cupation classification from the Common Crawl corpus. We report the overall accuracy of the task as well as the accuracy breakdown based on gender. To quantify gender bias, we compute the difference in true positive rates (TPR) between genders across various occupations, denoted as $\text{GAP}_g^{\text{TPR}}$ and defined as follows:

$$\text{GAP}_g^{\text{TPR}} = |\text{TPR}_g - \text{TPR}_{\sim g}|, \qquad (3)$$

where $\text{TPR}_g$ represents the proportion of individuals correctly predicted given their gender $g$, and $g$ and $\sim g$ are binary genders. Following (Romanov et al., 2019; Ravfogel et al., 2020; He et al., 2022), we also calculate the root mean square of the per-occupation TPR gender gap $\text{GAP}_{g,o}^{\text{TPR}}$ over all occupations $o$:

$$\text{GAP}_g^{\text{RMS}} = \sqrt{\frac{1}{|O|} \sum_{o \in O} \left(\text{GAP}_{g,o}^{\text{TPR}}\right)^2}. \qquad (4)$$

A value closer to 0 indicates a lower degree of bias.

## 4.2 Datasets and Setup

**STS-B and SNLI.** We fine-tune the models on SNLI and STS-B training sets and pick the one that performs best on the validation set and then evaluate bias using Bias-STS-B and Bias-NLI, respectively. **Bias-in-Bios.** We use the same data as Ravfogel et al. (2020) which contains 393,423 biographies and 28 profession classes. We split train/validation/test by 65/25/10 following (De-Arteaga et al., 2019; Ravfogel et al., 2020). The dataset statistics are shown in Table 1. For SNLI and Bias-in-Bios, we report accuracy over classification while we report the Pearson and Spearman correlations for STS-B.

## 4.3 Baseline Models

We compare Co²PT with six upstream debiased models fine-tuned on the downstream tasks, and three baselines fine-tune or prompt-tune BERT models on the downstream tasks: **ZariCDA** (Webster et al., 2020) is pre-trained from scratch over counterfactual data augmented from English Wikipedia and **ZariDO** (Webster et al., 2020) is additionally pre-trained with increased dropout rate.

| | Train | Validation | Bias-Test |
|---|---|---|---|
| STS-B | 5,749 | 1,500 | 16,980 |
| SNLI | 550,152 | 10,000 | 1,936,512 |
| Bias-in-Bios | 255,710 | 39,369 | 98,344 |

Table 1: Dataset statistics.

**ADELE** and its variant **ADELE-TA** (Lauscher et al., 2021) inject and train debiased adapters via masked language modeling on the counterfactually augmented corpus. **Context-Debias** (Kaneko and Bollegala, 2021) is fine-tuned via an inner-product loss and squared distance loss. **Auto-Debias** (Guo et al., 2022) uses a beam search to search for biased prompts and then uses these biased prompts to fine-tune PLMs by minimizing the disagreement between predicted [MASK] token distributions. **MA-BEL** (He et al., 2022) is additionally pre-trained on all entailment pairs that contain gendered terms from SNLI and MNLI data with a contrastive loss, an alignment loss, and an optional masked language modeling loss. **BERT** (Devlin et al., 2019) is a fine-tuned BERT model on the downstream tasks while **BERT+CDA** is a fine-tuned BERT model on the counterfactually augmented data from the training sets. **PT** (Liu et al., 2022) adds continuous prompts to each layer of the models and then tunes the prompts on downstream tasks. The backbone models for ZariCDA and ZariDO are `BERT-large-uncased` whereas other baselines are `BERT-base-uncased` (Devlin et al., 2019).[1]

## 4.4 Implementation Details

For fine-tuning debiased baselines and vanilla BERT, we use the models released by the authors. We set the max length of the sentence to be 128, the learning rate to be $2e-5$, the batch size to be 64, and train for 10 epochs. For AT+CDA, learning rate is 2e-5 and batch size is 128. For PT and Co²PT, the backbone models are `bert-base-uncased` with the learning rate set to $1e-2$ and the prompt length set to 20 and are trained for 30 epochs. The batch size is set to 32. For hyperparameters $\tau$ and $\alpha$ in Equation 1 and 2, we set $\tau = 0.05$ and $\alpha = 1.0$. All experiments are run on a single NVIDIA RTX A5000 24GB GPU. For each run, we save the model that performs the best on the development set and evaluate it on extrinsic benchmarks. We report the average results across three runs. All code and data are available at `https://github.com/dongxiangjue/Co2PT`; additional experimental details and standard deviations are in Appendix A and Appendix C, respectively.

---

[1]We keep the results of these two `BERT-large-uncased` checkpoints since Lauscher et al. (2021) also compares these methods with their `BERT-base-uncased` method.

# 5 Debiasing Effectiveness (RQ1)

We now investigate the effectiveness of $Co^2PT$ in mitigating bias on three extrinsic bias benchmarks.

**Bias-STS-B.** First, we focus on the Bias-STS-B benchmark. As Table 2 indicates, $Co^2PT$ shows the lowest bias scores across all metrics and achieves similar model performance on downstream tasks as the other debiased baselines. We observe that some debiased models exhibit higher bias scores than the original BERT, indicating that debiased language models can relearn biases during fine-tuning on downstream tasks. For example, Auto-Debias, one of the state-of-the-art debiased models, demonstrates strong fairness on intrinsic benchmarks, such as SEAT, scores 0.312 in average absolute difference, showing a higher bias level than the original BERT and most of the other baselines. On the other hand, MABEL, which shows strong performance in debiasing downstream tasks, achieves a competitive score of 0.081. Furthermore, compared to fine-tuning the original BERT model on the STS-B training set, PT results in a higher bias score of 0.321 compared to 0.282. This suggests that while fine-tuning only the number of prompt tokens may be parameter efficient, it can result in increased bias due to the presence of an unbalanced dataset. For $Co^2PT$, we observe a significant reduction in the bias score with the average absolute difference decreasing from 0.321 to 0.058, from 0.749 to 0.167 when the difference exceeds 0.1, and from 0.369 to 0.005 when the difference exceeds 0.3. These findings indicate a substantial improvement in the ability to mitigate bias.

**Bias-NLI.** Next, we focus on the Bias-NLI extrinsic benchmark shown in Table 3. BERT+CDA, Context-Debias, and MABEL achieve lower bias

| Model | NN↑ | FN↑ | T:0.5↑ | T:0.7↑ | Acc. |
|---|---|---|---|---|---|
| BERT | 0.824 | 0.868 | 0.867 | 0.811 | 0.909 |
| BERT+CDA | 0.873 | 0.942 | 0.941 | 0.894 | 0.905 |
| ZariCDA* | 0.786 | 0.828 | 0.826 | 0.765 | 0.912 |
| ZariDO* | 0.747 | 0.782 | 0.779 | 0.711 | 0.913 |
| Context-Debias | 0.873 | 0.919 | 0.919 | 0.877 | 0.910 |
| ADELE-TA† | 0.504 | 0.557 | - | - | 0.813 |
| AT+CDA‡ | 0.659 | 0.799 | 0.758 | 0.510 | 0.849 |
| Auto-Debias | 0.813 | 0.849 | 0.848 | 0.795 | 0.908 |
| MABEL | 0.853 | 0.892 | 0.891 | 0.846 | 0.911 |
| PT | 0.741 | 0.812 | 0.808 | 0.729 | 0.898 |
| $Co^2PT$ (ours) | **0.877** | **0.965** | **0.962** | **0.905** | 0.886 |

Table 3: Evaluation on Bias-NLI. †: results are fine-tuned on MNLI and reported from the ADELE-TA model in the original paper; ‡: adapter tuning on counterfactually augmented data; *: backbone model is `BERT-large-uncased`. Other baselines are fine-tuned on SNLI.

scores than the original BERT across all metrics while the other baseline methods amplify biases during the fine-tuning. Similarly, Auto-Debias performs well on the SEAT benchmark but experiences an increase in bias when applied to downstream tasks, mirroring the trend observed in the Bias-STS-B extrinsic benchmark. Moreover, ADELE, another parameter-efficient method, performs poorly in both bias mitigation and model accuracy. Similar to Bias-STS-B extrinsic benchmark, PT amplifies biases during the tuning process, resulting in a decline in the NN score from 0.824 to 0.741 and the FN score from 0.868 to 0.812. By employing $Co^2PT$, we observe significant improvements with the NN score rising to 0.877 (from 0.741) and the FN score reaching 0.965 (from 0.812), indicating the effectiveness of $Co^2PT$ on bias mitigation.

**Bias-in-Bios.** Next we show the performance on the Bias-in-Bios benchmark in Table 4. Among all the baselines, the ZariCDA achieves the lowest $GAP^{TPR}$ score of 2.667 while the BERT+CDA

| Model | Diff.↓ | $\tau$:0.1↓ | $\tau$:0.3↓ | Pear. / Spear. |
|---|---|---|---|---|
| BERT | 0.282 | 0.867 | 0.417 | 0.883 / 0.879 |
| BERT+CDA | 0.131 | 0.511 | 0.080 | 0.885 / 0.881 |
| ZariCDA* | 0.112 | 0.445 | 0.048 | 0.892 / 0.889 |
| ZariDO* | 0.347 | 0.922 | 0.585 | 0.880 / 0.878 |
| ADELE† | 0.121 | - | - | 0.889 / - |
| Context-Debias | 0.332 | 0.916 | 0.539 | 0.879 / 0.876 |
| Auto-Debias | 0.312 | 0.902 | 0.502 | 0.884 / 0.880 |
| MABEL | 0.066 | 0.204 | 0.013 | 0.889 / 0.885 |
| PT | 0.321 | 0.749 | 0.369 | 0.889 / 0.885 |
| $Co^2PT$ (ours) | **0.058** | **0.167** | **0.005** | 0.884 / 0.880 |

Table 2: Evaluation on Bias-STS-B. †: results are reported from the ADELE model in the original paper; *: backbone model is `BERT-large-uncased`.

| Model | $GAP^{TPR}$↓ | $GAP^{RMS}$↓ | Acc. |
|---|---|---|---|
| BERT | 2.822 | 0.119 | 0.830 |
| BERT+CDA | 2.758 | **0.113** | 0.841 |
| ZariCDA* | 2.667 | 0.135 | 0.848 |
| ZariDO* | 2.720 | 0.135 | 0.847 |
| Context-Debias | 3.031 | 0.119 | 0.831 |
| Auto-Debias | 2.690 | 0.125 | 0.831 |
| MABEL | 2.839 | 0.126 | 0.826 |
| PT | 3.171 | 0.129 | 0.820 |
| $Co^2PT$ (ours) | **2.537** | 0.123 | 0.824 |

Table 4: Evaluation on Bias-in-Bios. *: backbone model is `BERT-large-uncased`. The results of ADELE on this benchmark are not reported in the original paper.

achieves the lowest GAP$^{\text{RMS}}$ score of 0.113. Furthermore, PT exacerbates bias and results in an increase in the GAP$^{\text{TPR}}$ score from 2.822 to 3.171 and GAP$^{\text{RMS}}$ from 0.119 to 0.129. In contrast, Co$^2$PT reduces the GAP$^{\text{TPR}}$ score from 3.171 to 2.537 and GAP$^{\text{RMS}}$ from 0.129 to 0.123, demonstrating its effectiveness in mitigating bias in the occupation classification task.

# 6 Integrating Co$^2$PT with Existing Debiased Models (RQ2)

One benefit of a prompt-then-finetune model like Co$^2$PT is that it can be easily integrated with existing upstream debiasing methods. Here we investigate the applicability of Co$^2$PT to three existing debiased models to bridge the gap in utilizing upstream debiased models for downstream tasks. Based on the comparison results – before and after applying Co$^2$PT – shown in Table 5, Co$^2$PT significantly reduces the bias scores for Context-Debias and Auto-Debias: 0.088 versus 0.332, and 0.068 versus 0.312, respectively. For MABEL, which achieves low bias scores, there is no significant effect on the bias score. Additionally, Co$^2$PT improves the model performance on the downstream tasks. These results clearly demonstrate the effectiveness of integrating Co$^2$PT into established debiased models for downstream tasks. This ability enables the existing debiased models to achieve strong performance on downstream tasks while simultaneously maintaining a low bias level.

| Model | Diff.↓ | $\tau$:0.1↓ | $\tau$:0.3↓ | Pear. / Spear. |
|---|---|---|---|---|
| Context-Debias | 0.332 | 0.916 | 0.539 | 0.879 / 0.876 |
| + Co$^2$PT | **0.088** | **0.361** | **0.010** | 0.885 / 0.881 |
| Auto-Debias | 0.312 | 0.902 | 0.502 | 0.884 / 0.880 |
| + Co$^2$PT | **0.068** | **0.231** | **0.005** | 0.883 / 0.878 |
| MABEL | **0.066** | **0.204** | 0.013 | 0.889 / 0.885 |
| + Co$^2$PT | 0.068 | 0.228 | **0.005** | 0.892 / 0.889 |

Table 5: Performance of integrating Co$^2$PT with debiased models on Bias-STS-B.

# 7 Impact of Design (RQ3)

We perform an extensive ablation study to show how different components affect Co$^2$PT in Table 6. We use Bias-STS-B as the representative task for computational efficiency.

**Impact of counterfactual module.** First, we perform counterfactual data augmentation on the training data containing bias-attribute terms. Then we conduct prompt tuning only on these augmented pairs (denoted as **PT+CDA**). PT+CDA reduces

the bias score from 0.321 in PT to 0.291, showing the effectiveness of the straightforward counterfactual data augmentation approach. However, the improvement is less than Co$^2$PT, implying the necessity of the contrastive learning module.

**Impact of contrastive module.** To investigate the impact of the contrastive module, instead of employing constructed counterfactual sentence pairs as positive pairs for contrastive loss, we use unsupervised contrastive loss by encoding the same input twice and get two embeddings with different dropout masks $z$, $z'$ (Gao et al., 2021). Then the contrastive objective becomes:

$$\mathcal{L}_{scl} = -\log \frac{e^{\text{sim}(\mathbf{p}\oplus\mathbf{h}_i^{z_i},\mathbf{p}\oplus\mathbf{h}_i^{z'_i})/\tau}}{\sum_{j=1}^{N} e^{\text{sim}(\mathbf{p}\oplus\mathbf{h}_i^{z_i},\mathbf{p}\oplus\mathbf{h}_j^{z'_j})/\tau}}, \quad (5)$$

which is optimized with the prompt tuning loss $\mathcal{L}_{pt}$ together (denoted as **PT+SCL**). PT+SCL surpasses both PT and PT+CDA by achieving a large reduction in bias score to 0.161, demonstrating the effectiveness of the contrastive module.

**Impact of adding contrastive loss for non-augmented inputs.** In Co$^2$PT, we only consider counterfactually augmented pairs in the contrastive module. To explore the necessity of including inputs without demographic terms in the contrastive module, we incorporate unsupervised contrastive loss for non-augmented input like Equation 5 as $\mathcal{L}'_{scl}$ and tuned with contrastive loss $\mathcal{L}_{cl}$ for counterfactually augmented pairs (Equation 1) and $\mathcal{L}'_{pt}$ (denoted as **Co$^2$PT+SCL$_n$**). The bias score of 0.117 achieved by Co$^2$PT+SCL$_n$ is higher than Co$^2$PT and this indicates that incorporating a contrastive loss for non-augmented inputs in the training set is unnecessary.

**Compare Co$^2$PT with task-agnostic counterfactual pairs.** To investigate whether integrating task-agnostic neutral entailment pairs can benefit debiasing on the task, we use 142,158 gender-balanced entailment pairs augmented from SNLI and MNLI

| Model | Diff.↓ | $\tau$:0.1↓ | $\tau$:0.3↓ | Pear. / Spear. |
|---|---|---|---|---|
| PT | 0.321 | 0.749 | 0.369 | 0.889 / 0.885 |
| PT+CDA | 0.291 | 0.747 | 0.351 | 0.890 / 0.886 |
| PT+SCL | 0.161 | 0.548 | 0.133 | 0.883 / 0.878 |
| Co$^2$PT+SCL$_n$ | 0.117 | 0.467 | 0.056 | 0.884 / 0.878 |
| PT+NLI+CL | 0.080 | 0.280 | 0.022 | 0.881 / 0.876 |
| PT+NLI+CL$_p$ | 0.207 | 0.687 | 0.222 | 0.884 / 0.881 |
| PT+CDA+CL$_p$ | 0.271 | 0.725 | 0.338 | 0.886 / 0.883 |
| Co$^2$PT (PT+CDA+CL) | **0.058** | **0.167** | **0.005** | 0.884 / 0.880 |

Table 6: Impact of different components.

datasets in He et al. (2022) as task-agnostic entailment pairs for STS-B task instead of using the task-specific counterfactual pairs augmented from the training set (denoted as **PT+NLI+CL**). We notice that although PT+NLI+CL does not outperform $Co^2PT$, it shows a strong ability to mitigate bias compared to other baseline methods. Thus, when working on a moderate amount of training data, it is better to use counterfactually augmented pairs from the training data.

**Compare $Co^2PT$ with other contrastive objective.** For sentence-pair classification tasks, we also explore the contrastive loss that encourages the inter-association of entailment pairs (He et al., 2022). For the original input pair $(s_{i1}, s_{i2})$ and its augmented pair $(s'_{i_1}, s'_{i_2})$, $s_{i1}$ and $s_{i2}$ are treated as positive pairs while $s_{i1}$ and $s'_{i2}$ and other in-batch $s_{j2}$ are negatives, and vice versa (denoted as **CL$_p$**). When using task-specific counterfactual pairs, **PT+CDA+CL$_p$** decreases the bias score to 0.271. Similarly, using task-agnostic counterfactual pairs, **PT+NLI+CL$_p$** also reduces the bias score to 0.271. However, the bias mitigation effect is not as significant as that achieved by $Co^2PT$, which indicates the effectiveness of the contrastive module in $Co^2PT$.

## 8 Impact of Hyperparameters (RQ4)

Finally, we investigate the impact of three hyperparameters: (i) the continuous prompt length; (ii) the temperature $\tau$ of contrastive loss $\mathcal{L}_{cl}$; and (iii) the coefficient $\alpha$ of total learning objective $\mathcal{L}$.

**Impact of prompt length.** First, we experiment with the prompt length varying in $\{10, 20, 50\}$, as illustrated in Figure 2. Generally speaking, with more tunable prompt parameters, the model performs better on downstream tasks. In addition, when the prompt length is 10, $Co^2PT$ shows a higher increase in bias score compared to the prompt length of 20 and 50. This indicates that a

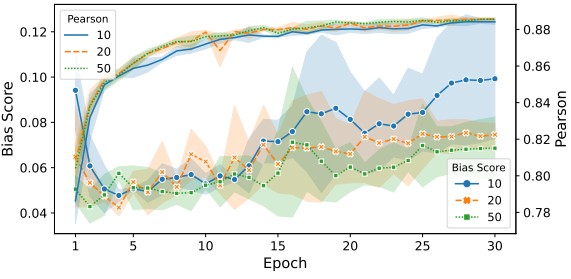

Figure 2: Impact of prompt length.

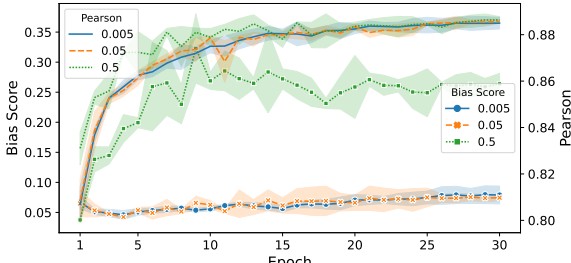

Figure 3: $\tau$ in $\{0.005, 0.05, 0.5\}$ while $\alpha = 1.0$.

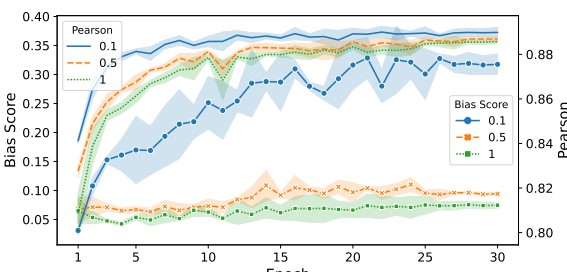

Figure 4: $\alpha$ in $\{0.1, 0.5, 1.0\}$ while $\tau = 0.05$.

larger prompt length enables the model to achieve better model performance on downstream tasks more rapidly while still maintaining a lower bias score. However, it is important to consider that using larger prompt lengths means tuning more parameters, thus posing a trade-off.

**Impact of $\tau$.** Then, we vary temperature $\tau$ in $\{0.005, 0.05, 0.5\}$. Figure 3 shows close significant bias mitigation effects when $\tau$ is set to 0.005 and 0.05 while exhibiting less effectiveness when $\tau$ is 0.5. This observation implies that a higher temperature value corresponds to less weight of the cosine similarity calculation, resulting in decreased effectiveness in bias mitigation.

**Impact of $\alpha$.** Last, we study the impact of coefficient $\alpha$ and vary the value in $\{0.1, 0.5, 1.0\}$. Figure 4 emphasizes that reducing the value of $\alpha$ at a constant $\tau$, thus assigning less weight to the contrastive module, leads to decreased bias mitigation effects. This analysis underscores the importance of carefully selecting appropriate hyperparameters.

## 9 Conclusion and Future Work

We propose $Co^2PT$, an efficient and effective debiasing method for mitigating bias in downstream tasks. We evaluate its effectiveness on bias mitigation and applicability to existing debiased upstream models, and investigate how the design of each component and the selection of hyperparameters impact both its bias reduction capabilities and downstream task performance.

**Mitigating non-gender and intersectional bias.**
Mitigating non-gender biases is challenging as some debiasing methods work well on reducing gender biases but show poor generalization capabilities in addressing biases beyond gender (Meade et al., 2022). Without re-training the model, Co$^2$PT is flexible to apply in order to mitigate different bias types in downstream applications. One can train different debiasing prompts to tackle different bias dimensions such as gender, race, and religion. Furthermore, these debiasing prompts can be applied to mitigate intersectional bias by simply combining the corresponding prompts in downstream tasks.

## Limitations

While this work primarily addresses bias in English, we acknowledge the presence of more complicated bias cases in other languages. Therefore, future exploration of existing methods or the development of new techniques to mitigate bias in other languages would be valuable. Furthermore, despite the efficiency and comparable performance of deep prompt tuning compared to fine-tuning, it still underperforms fine-tuning on certain datasets when the model size is small. This will also limit the model performance of our method.

## Ethics Statement

In this work, when investigating gender bias in pre-trained language models, we focus on the binary definition of gender as the targeted attribute of discrimination. However, it is important to acknowledge that future research should also consider non-binary genders and other multi-class scenarios to comprehensively address bias.

## Acknowledgements

We thank Nicholas Meade for the initial discussion and anonymous reviewers for valuable feedback.

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

# Appendix

## A More Implementation Details

### A.1 Dataset and Extrinsic Bias Benchmarks

STS-B and SNLI datasets are from the Hugging Face Datasets library (Lhoest et al., 2021).[2] We use the same Bias-in-Bios dataset used in Ravfogel et al. (2020).[3], the Bias-STS-B used in Lauscher et al. (2021) and the Bias-NLI used in He et al. (2022). The code used to assess Bias-STS-B is modified from Lauscher et al. (2021), while the code for evaluating Bias-NLI and Bias-in-Bios is adapted from He et al. (2022). Different from the code employed in He et al. (2022), we conduct our evaluation on the entire Bias-NLI dataset rather than just the top 10% of it.

### A.2 Debiased Baselines

Please refer to the footnotes here for the source of released debiased models – ZariCDA, ZariDO[4] (Webster et al., 2020), Context-Debias[5] (Kaneko and Bollegala, 2021), Auto-Debias[6] (Guo et al., 2022), MABEL[7] (He et al., 2022) – used on downstream tasks.

### A.3 Implementation Details

We use BERT-base-uncased in our experiments. For the single-sentence classification task, we prepend the [CLS] token before the input sentences and feed it into the BERT model to get the embedding of the [CLS] token as the sentence representation. For the sentence-pair classification task, e.g., SNLI task, we prepend [CLS] before the premise $x$ and [SEP] token to separate premise $x$ and hypothesis $y$ and feed it into the BERT model to get the embedding of the [CLS] token as the sentence representation. Fine-tuning and prompt tuning code rely on the Huggingface implementation.[8]

---

[2] https://github.com/huggingface/datasets. Apache License 2.0.

[3] The data is downloaded through https://github.com/shauli-ravfogel/nullspace_projection/blob/master/download_data.sh MIT License.

[4] The checkpoints of these two models are from https://github.com/google-research-datasets/Zari. Apache-2.0 license.

[5] The checkpoints are from https://github.com/kanekomasahiro/context-debias. MIT license.

[6] The checkpoints are from https://github.com/Irenehere/Auto-Debias.

[7] The checkpoints are from https://huggingface.co/princeton-nlp/mabel-bert-base-uncased MIT License.

[8] https://github.com/huggingface

## B More Ablation Studies

**Pooling method.** Besides using pooled output, we also conduct experiments that use the average token representation from the model's last hidden state as sentence representation, shown in Table 7. However, upon analyzing the results, $Co^2PT_{avg.}$ is considerably more challenging for the prompts to acquire the debiasing capability when using the average token representation compared to when utilizing the CLS token as the sentence representation.

| Model | Diff.↓ | $\tau$:0.1↓ | $\tau$:0.3↓ | Pear. / Spear. |
|---|---|---|---|---|
| PT | 0.321 | 0.749 | 0.369 | 0.889 / 0.885 |
| $Co^2PT_{avg.}$ | 0.390 | 0.823 | 0.497 | 0.891 / 0.886 |
| $Co^2PT_{cls}$ (ours) | **0.058** | **0.167** | **0.005** | 0.884 / 0.880 |

Table 7: More ablation results.

## C Standard Deviation of Results

The standard deviation of evaluation results on extrinsic bias benchmarks – Bias-STS-B, Bias-NLI, and Bias-in-Bios – are presented in Tables 8 to 10, respectively. In addition, the results of integrating $Co^2PT$ with existing debiased models and ablation study are shown in Table 11 and Table 12. These results indicate that $Co^2PT$ consistently performs well with relatively low variability, demonstrating its effectiveness and reliability.

| Model | Diff.↓ | $\tau$:0.1↓ | $\tau$:0.3↓ | Pear. / Spear. |
|---|---|---|---|---|
| BERT | 0.018 | 0.024 | 0.056 | 0.002 / 0.001 |
| BERT+CDA | 0.020 | 0.058 | 0.042 | 0.001 / 0.000 |
| ZariCDA⋆ | 0.030 | 0.146 | 0.031 | 0.004 / 0.003 |
| ZariDO⋆ | 0.020 | 0.011 | 0.044 | 0.003 / 0.003 |
| Context-Debias | 0.042 | 0.052 | 0.098 | 0.005 / 0.005 |
| Auto-Debias | 0.006 | 0.018 | 0.020 | 0.002 / 0.001 |
| MABEL | 0.011 | 0.046 | 0.013 | 0.004 / 0.004 |
| PT | 0.018 | 0.016 | 0.024 | 0.001 / 0.001 |
| $Co^2PT$ (ours) | 0.009 | 0.056 | 0.001 | 0.001 / 0.000 |

Table 8: Results standard deviation on Bias-STS-B. ⋆: the backbone models are BERT-large-uncased.

## D Visualization of Bias Mitigation Effects along Epochs

We visualize the changes of bias scores along epochs in Figure 5. The bias score of PT keeps increasing as the Pearson score increases, while $Co^2PT$ consistently maintains a low bias score, which indicates the effectiveness of $Co^2PT$ on bias mitigation.

| Model | NN↑ | FN↑ | T:0.5↑ | T:0.7↑ | Acc. |
|---|---|---|---|---|---|
| BERT | 0.027 | 0.042 | 0.041 | 0.036 | 0.002 |
| BERT+CDA | 0.012 | 0.010 | 0.010 | 0.002 | 0.003 |
| ZariCDA⋆ | 0.082 | 0.081 | 0.082 | 0.097 | 0.001 |
| ZariDO⋆ | 0.078 | 0.083 | 0.083 | 0.093 | 0.001 |
| Context-Debias | 0.030 | 0.046 | 0.046 | 0.050 | 0.001 |
| Auto-Debias | 0.010 | 0.019 | 0.019 | 0.012 | 0.002 |
| MABEL | 0.018 | 0.003 | 0.002 | 0.014 | 0.001 |
| PT | 0.023 | 0.032 | 0.031 | 0.028 | 0.001 |
| $Co^2PT$ (ours) | 0.016 | 0.021 | 0.022 | 0.027 | 0.004 |

Table 9: Results standard deviation on Bias-NLI. ⋆: the backbone models are `BERT-large-uncased`.

| Model | GAP$^{\text{TPR}}$↓ | GAP$^{\text{RMS}}$↓ | Acc. |
|---|---|---|---|
| BERT | 0.138 | 0.004 | 0.001 |
| BERT+CDA | 0.034 | 0.001 | 0.002 |
| ZariCDA⋆ | 0.069 | 0.006 | 0.000 |
| ZariDO⋆ | 0.074 | 0.010 | 0.003 |
| Context-Debias | 0.053 | 0.006 | 0.003 |
| Auto-Debias | 0.146 | 0.004 | 0.002 |
| MABEL | 0.114 | 0.006 | 0.001 |
| PT | 0.066 | 0.005 | 0.001 |
| $Co^2PT$ (ours) | 0.025 | 0.005 | 0.007 |

Table 10: Results standard deviation on Bias-in-Bios. ⋆: the backbone models are `BERT-large-uncased`.

| Model | Diff.↓ | $\tau$:0.1↓ | $\tau$:0.3↓ | Pear. / Spear. |
|---|---|---|---|---|
| Context-Debias | 0.042 | 0.052 | 0.098 | 0.005 / 0.005 |
| + $Co^2PT$ | 0.031 | 0.194 | 0.011 | 0.000 / 0.001 |
| Auto-Debias | 0.006 | 0.018 | 0.020 | 0.002 / 0.001 |
| + $Co^2PT$ | 0.008 | 0.037 | 0.005 | 0.002 / 0.002 |
| MABEL | 0.011 | 0.046 | 0.013 | 0.004 / 0.004 |
| + $Co^2PT$ | 0.021 | 0.103 | 0.023 | 0.000 / 0.000 |

Table 11: Performance of integrating $Co^2PT$ with debiased models on Bias-STS-B.

| Model | Diff.↓ | $\tau$:0.1↓ | $\tau$:0.3↓ | Pear. / Spear. |
|---|---|---|---|---|
| PT | 0.018 | 0.016 | 0.024 | 0.001 / 0.001 |
| PT+CDA | 0.008 | 0.008 | 0.007 | 0.000 / 0.000 |
| PT+SCL | 0.025 | 0.056 | 0.039 | 0.000 / 0.000 |
| $Co^2PT$+SCL$_n$ | 0.023 | 0.094 | 0.033 | 0.001 / 0.001 |
| PT+NLI+CL | 0.013 | 0.070 | 0.012 | 0.001 / 0.001 |
| PT+NLI+CL$_p$ | 0.025 | 0.042 | 0.053 | 0.001 / 0.001 |
| PT+CDA+CL$_p$ | 0.030 | 0.031 | 0.051 | 0.001 / 0.002 |
| $Co^2PT$ (avg.) | 0.032 | 0.022 | 0.040 | 0.001 / 0.000 |
| $Co^2PT$ (PT+CDA+CL) | 0.009 | 0.056 | 0.001 | 0.001 / 0.000 |

Table 12: Results standard deviation of ablation study.

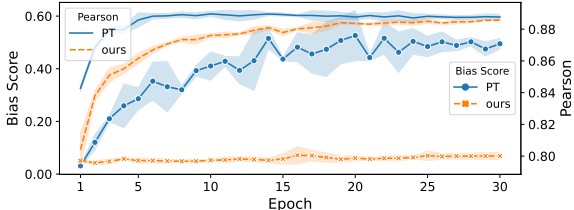

Figure 5: Visualization of bias mitigation effects and model performance along epochs.