# OpenReview forum: "Co$^2$PT: Mitigating Bias in Pre-trained Language Models through Counterfactual Contrastive Prompt Tuning"
_EMNLP/2023/Conference — EMNLP 2023 Findings_

### Official Review · Reviewer_ppym · 2023-07-21

**Soundness:** 4

**Excitement:**

3: Ambivalent: It has merits (e.g., it reports state-of-the-art results, the idea is nice), but there are key weaknesses (e.g., it describes incremental work), and it can significantly benefit from another round of revision. However, I won't object to accepting it if my co-reviewers champion it.

**Paper Topic And Main Contributions:**

This paper aims to improve the efficiency and effectiveness of mitigating gender bias on (extrinsic) downstream tasks. The authors propose to integrate a counterfactual contrastive loss when adapting pre-trained language models to downstream tasks via deep prompt tuning. Main contributions seem to be (i) the method of debiasing during adaptation to downstream tasks (or the so-called ''debias-while-prompt-tuning'') and (ii) adopting deep prompt tuning for bias mitigation. The authors demonstrated the effectiveness of their method on three extrinsic bias benchmarks: Bias-STS-B, Bias-NLI and Bias-in-Bios.

**Questions For The Authors:**

A. Based on my understanding, when fine-tuning the eight baseline methods on downstream tasks, you use the original training set, not the augmented balanced training set, right? Given this, I'm confused by the results in Table 3 that ADELE's debiasing performance is much worse than the vanilla BERT. This may indicate some problems in your implementation. Is this due to that ADELE is trained on MNLI? If so, I think you should re-train ADELE using the same training set as yours for a fair comparison.

B. In Table 6, you showed that PT+SCL can significantly reduce bias, which is achieved only by the contrastive loss of a sentence itself, without any bias terms telling the model what kind of bias (i.e., gender bias) it is to reduce. This seems a bit unreasonable to me. Could you explain more on this?

C. I'm also curious about how PT+CDA in Table 6 is implemented. Do you assume the augmented instance has the same label as the original instance, and optimize the classification loss on both instances?

D. You mentioned in lines 110-112 & 659-660 that different prompts can be combined to mitigate intersectional biases, but this is not obvious to me. Could you present a concrete method of combining different prompts? This is not a necessary problem for this paper, but would make the proposed method much more useful in practice.

**Reasons To Accept:**

The authors proposed a counterfactual contrastive loss and a prompt tuning-based method to mitigate gender bias in downstream tasks. The method is parameter-efficient and compatible to upstream debiasing techniques.

The authors conducted extensive ablation studies to justify their design, especially the counterfactual contrastive loss, and shared their codes for reproduction. The experimental setups such as reporting results from several runs, providing the validation details, etc., well support the transparency of the experiments.

**Reasons To Reject:**

Some of the main ideas in this paper are not novel. (1) The idea of debiasing during adaptation to downstream tasks is not novel. Early works like [1][2] train coreference resolution models on augmented balanced training sets to mitigate gender bias in coreference resolution, which is similar to this paper's idea. (2) There are also recent works using prefix tuning or prompt tuning for bias mitigation[3].

The debiasing baselines in this paper are a bit weak, which makes it unclear if the proposed method's performance is truly superior. E.g., (1) in Table 2, the reported results for ADELE is by finetuning all the parameters of a debiased language model, but in the ADELE paper[4], there's a more effective method called ADELE-TA, and I think the authors should compare to this baseline. (2) Besides, the proposed counterfactual contrastive loss is also applicable to fine-tuning or other parameter efficient methods like adapter tuning, and I think these highly related baselines should be compared with to justify the superiority of the authors' choice of prompt tuning.

[1] Jieyu Zhao, Tianlu Wang, Mark Yatskar, Vicente Ordonez, Kai-Wei Chang. Gender Bias in Coreference Resolution: Evaluation and Debiasing Methods. NAACL 2018.

[2] Jieyu Zhao, Tianlu Wang, Mark Yatskar, Ryan Cotterell, Vicente Ordonez, Kai-Wei Chang. Gender Bias in Contextualized Word Embeddings. NAACL 2019.

[3] Ke Yang, Charles Yu, Yi R. Fung, Manling Li, Heng Ji. ADEPT: A DEbiasing PrompT Framework. AAAI 2023. (arXiv Dec 22)

[4] Anne Lauscher, Tobias Lueken, and Goran Glavaš. Sustainable modular debiasing of language models. Findings of EMNLP 2021.

**Reproducibility:**

4: Could mostly reproduce the results, but there may be some variation because of sample variance or minor variations in their interpretation of the protocol or method.

**Reviewer Confidence:**

4: Quite sure. I tried to check the important points carefully. It's unlikely, though conceivable, that I missed something that should affect my ratings.

**Typos Grammar Style And Presentation Improvements:**

There are concurrent works [1][2] adopting prefix tuning or prompt tuning for bias mitigation in downstream tasks, so the authors may cite them and revise the writings in lines 115-117: " the proposed Co2 PT is the first to mitigate bias on downstream tasks through prompt tuning".

[1] Zhongbin Xie and Thomas Lukasiewicz. An Empirical Analysis of Parameter-Efficient Methods for Debiasing Pre-Trained Language Models. ACL 2023.

[2] Yingji Li, Mengnan Du, Xin Wang, and Ying Wang. Prompt Tuning Pushes Farther, Contrastive Learning Pulls Closer: A Two-Stage Approach to Mitigate Social Biases. ACL 2023.

---

> ### Author Rebuttal · Authors · 2023-08-29
>
> We appreciate your positive comments that our proposed method is parameter-efficient and compatible with upstream debiasing techniques and extensive ablation studies are conducted to justify the design. We do our best to address your concerns below:
>
> **Weakness 1: Some of the main ideas in this paper are not novel, e.g., augmenting balanced training sets and prompt tuning.**
>
> We agree that training data augmentation and prompt tuning have been explored in prior literature.
>
> However:
> * As we point out in our contributions and related work, Co$^2$PT achieves time and memory efficiency without requiring access to an external corpus or retraining the entire model.
> * Co$^2$PT demonstrates better bias mitigation performance on downstream tasks.
> * Co$^2$PT can be flexibly integrated with existing debiased language models, effectively bridging the gap between debiased upstream models and downstream tasks. It can not only preserve the original debiased sentence representations by tuning prompts while freezing all parameters of the upstream models but also avoid re-acquiring or amplifying biases on downstream tasks.
>
> **Weakness 2(a): In Table 2, the reported results for ADELE are by finetuning all the parameters of a debiased language model, but I think the authors should compare to the more effective method called ADELE-TA.**
>
> We report the ADELE-TA results on Bias-NLI because they were provided in their paper [1]. We report the ADELE results on Bias-STS-B because they were provided in their paper [1]. The results for ADELE-TA on Bias-STS-B, however, are not reported in their paper [1]. Hence, we do not report the ADELE-TA on Bias-STS-B in our work.
>
> We will update the note "results are reported from the original paper" to "results are reported from the ADELE model in the original paper” in Table 2 and “results are reported from the ADELE-TA model in the original paper" in Table 3 to make it clearer.
>
> **Weakness 2(b): The proposed counterfactual contrastive loss is also applicable to fine-tuning or other parameter-efficient methods like adapter tuning, and I think these highly related baselines should be compared to justify the superiority of the authors' choice of prompt tuning.**
>
> In our work, we not only aim to mitigate bias on downstream tasks, but also we aim to preserve the performance capabilities of our backbone model. Based on previous work [2], the deep prompt tuning framework shows better performance than other existing prompt tuning approaches (e.g., P-tuning (Liu et al., 2021), PromptTuning (Lester et al., 2021)., Prefix Tuning (Li and Liang, 2021), Soft Prompts (Qin and Eisner, 2021)). Deep prompt tuning incorporates continuous prompts in every layer of the PLM instead of other prompt tuning methods, like only adding continuous prompts in the input embedding layer, which have more tunable task-specific parameters to enhance per-task capacity while maintaining parameter efficiency (Li and Liang, 2021; Liu et al., 2022). Thus, we choose to use deep prompt tuning.
>
> **Question A: ADELE's debiasing performance is much worse than the vanilla BERT.**
>
> We report ADELE-TA's results from their paper [1] showing the same findings as the recently published paper MABEL [3], where the TN and  FN of original BERT are 0.799 and 0.879--much better than ADELE.
>
> **Question B: PT+SCL can reduce bias, which is achieved only by the contrastive loss of a sentence itself., without any bias terms telling the model what kind of bias it is to reduce.**
>
> Applying contrastive loss to the sentence itself by encoding the same input twice and getting two embeddings with different dropout masks as noise, dropout acts as minimal “data augmentation” of hidden representations. It improves uniformity and the expressiveness of the representations [4], which we think is the reason why it also leads to bias reduction. While PT+SCL reduces bias, telling the model the biased terms (Co$^2$PT) reduces the bias further. This shows that emphasizing the biased signal to the model results in more robust debiasing.
>
> **Question C: I'm also curious about how PT+CDA in Table 6 is implemented. Do you assume the augmented instance has the same label as the original instance, and optimize the classification loss on both instances?**
>
> Yes. As stated in the Counterfactual Pairs Construction paragraph (line 206 to line 228) we replace the bias-attribute terms in the training data with the terms in the opposite bias direction. Then the augmented instance has the same label as the original instance and optimizes the classification loss.
>
> **Question D:  You mentioned that different prompts can be combined to mitigate intersectional biases, but this is not obvious to me. Could you present a concrete method of combining different prompts? This is not a necessary problem for this paper, but would make the proposed method much more useful in practice.**
>
> One intuitive way is to apply Co$^2$PT to train different continuous prompts for gender, religion, and race separately, and then concatenate the prompts to see whether it can mitigate these concatenated intersectional biases on downstream tasks. For example, a prompt trained to mitigate gender-related bias (e.g., women) and one to mitigate race-related bias (e.g., black), could be concatenated (to form one prompt from these two prompts) to mitigate intersectional bias (e.g., black woman).
>
> Another way is to apply Co$^2$PT with intersectional data augmentation to get the prompts to mitigate intersectional biases. For example, in previous gender-related augmentation, we replace “man” with “woman” in the data, then for intersectional-related augmentation, we can replace “man” with “black woman”.
>
> These are our initial ideas for future work. Feel free to collaborate if you are interested in the future work.
>
> **Add new citations in lines 115-117**
>
> Thank you for pointing out the newly published papers in ACL 2023. We will add the citations as you suggested.
>
> **References:**
>
> [1] Anne Lauscher, Tobias Lueken, and Goran Glavaš. 2021. Sustainable Modular Debiasing of Language Models. In Findings of the Association for Computational Linguistics: EMNLP 2021, pages 4782–4797, Punta Cana, Dominican Republic. Association for Computational Linguistics.
>
> [2] Xiao Liu, Kaixuan Ji, Yicheng Fu, Weng Tam, Zhengxiao Du, Zhilin Yang, and Jie Tang. 2022. P-tuning: Prompt tuning can be comparable to fine-tuning across scales and tasks. In Proceedings of the 60th Annual Meeting of the Association for Computational Linguistics (Volume 2: Short Papers), pages 61–68, Dublin, Ireland. Association for Computational Linguistics.
>
> [3] Jacqueline He, Mengzhou Xia, Christiane Fellbaum, and Danqi Chen. 2022. MABEL: Attenuating Gender Bias using Textual Entailment Data. In Proceedings of the 2022 Conference on Empirical Methods in Natural Language Processing, pages 9681–9702, Abu Dhabi, United Arab Emirates. Association for Computational Linguistics.
>
> [4] Tianyu Gao, Xingcheng Yao, and Danqi Chen. 2021. SimCSE: Simple Contrastive Learning of Sentence Embeddings. In Proceedings of the 2021 Conference on Empirical Methods in Natural Language Processing, pages 6894–6910, Online and Punta Cana, Dominican Republic. Association for Computational Linguistics.

---

### Official Review · Reviewer_vAfa · 2023-08-02

**Soundness:** 3

**Excitement:**

2: Mediocre: This paper makes marginal contributions (vs non-contemporaneous work), so I would rather not see it in the conference.

**Paper Topic And Main Contributions:**

The main contribution of the paper is to propose a method called Co2PT (Counterfactual Contrastive Prompt Tuning) to mitigate social bias on pre-trained language models (PLMs) by de-biasing in downstream tasks via prompt tuning.The Co2PT method first adds tunable continuous prompts to the pre-trained model by adding tunable continuous prompts, and then incorporates the debiasing capability into the prompt parameters by constructing counterfactual pairs from the training data and optimizing the prompt parameters by comparing the objectives while keeping the PLM parameters frozen. The method saves computational resources and memory during the debiasing process and can effectively mitigate the bias amplified by the model during prompt tuning in the downstream task.

**Questions For The Authors:**

Q1. On lines 267-271, for a context-embedded PLM, how do you obtain a contextual representation of tokens (e.g., e^[CLS])?

Q2. Explain the motivation for using prompts and what benefits it brings to Co2PT.

Q3. The debiasing of models often affects the performance of the model because counterfactual pairs are different from the real corpus, how did the authors balance the debiasing of the model with the performance?

**Reasons To Accept:**

A1. The authors propose a novel method for debiasing that is efficient and effective, and demonstrate its effectiveness through experiments.

A2. The paper is well-written and clearly presents the authors' contributions and findings.

**Reasons To Reject:**

R1. It seems reasonable to construct counterfactual pairs, and then fine-tune and debias the model, but the specific role of prompts still lacks explainability.

R2. It is not possible to clarify the effect of prompt on debiasing methods, such as ablation experiments.

R3. The authors claim that Co2PT does not require external corpora, but using prompts and counterfactual pairs is not necessarily a simpler resource than external corpora.

**Reproducibility:**

4: Could mostly reproduce the results, but there may be some variation because of sample variance or minor variations in their interpretation of the protocol or method.

**Reviewer Confidence:**

4: Quite sure. I tried to check the important points carefully. It's unlikely, though conceivable, that I missed something that should affect my ratings.

---

> ### Author Rebuttal · Authors · 2023-08-29
>
> We appreciate your positive comments that our proposed method is novel, efficient, and effective and the paper is well-written and clear. We turn now to each of your points:
>
> **R1: The specific role of prompts lacks explainability.**
>
> There are two main reasons for our use of prompts:
>
> 1) **Efficiency in tuning.** In this paper, we adopt deep prompt tuning as the backbone framework of Co$^2$PT. We incorporate continuous prompts as prefix tokens in every layer of the PLM. We should note that Co$^2$PT uses continuous prompts, which perform prompting directly in the embedding space of the model instead of discrete prompts, e.g., natural language phrases. As the goal of constructing prompts is to find a method that allows an LM to effectively perform a task rather than being for human consumption, it is not necessary to limit the prompt to human-interpretable natural language. Continuous prompts have their own parameters that can be tuned based on training data from the downstream task [1]. Thus, instead of fine-tuning the whole parameters of a pre-trained model, we only tune 0.1%-3% of parameters, which substantially reduces training time memory cost and per-task storage cost [2].
>
> Co$^2$PT is efficient especially i) when the LLMs have **large sizes of parameters** (e.g., 11B), which is extremely expensive to be tuned; ii) the models are **black-box** and the users can not get access to the model parameters.
>
> 2) **Effectiveness on bias mitigation.** Applying the counterfactual contrastive prompt tuning, we aim to store the debiasing knowledge in the prompt parameters. This approach not only retains the knowledge within the original parameters of PLMs but is also flexible and adaptable to different downstream tasks. In addition, it can be flexibly integrated with different existing debiased models for bias mitigation on downstream tasks.
>
> **R2: It is not possible to clarify the effect of prompt on debiasing methods, such as ablation experiments.**
>
> 1) In our ablation experiments in Section 8 (Impact of Hyperparameters), we experiment with the effectiveness of continuous prompts in different lengths varying in {10, 20, 50}. Generally speaking, with more tunable prompt parameters, the model performs better on downstream tasks. This indicates that a larger prompt length enables the model to achieve better model performance on downstream tasks more rapidly while still maintaining a lower bias score.
> 2) In addition, when the prompt length is set, assigning less weight to the contrastive module leads to decreased bias mitigation effects, as shown in Figure 4.
>
> **R3: Using prompts and counterfactual pairs is not necessarily a simpler resource than external corpora.**
>
> 1) Not every downstream task has a corresponding appropriate external corpora, and using the same (perhaps non-corresponding) external corpora for different downstream tasks cannot guarantee the effectiveness of bias mitigation due to issues such as instance-related biases and label-related biases in the downstream datasets. For example: debiasing an upstream model using balanced Wikipedia data may not mitigate the end bias on a sentiment classification task. Further, bias can still be re-acquired or amplified during fine-tuning on sentiment classification tasks.
> 2) Finding the proper external corpora for different downstream tasks is time-consuming and laborious. Thus, it can be more convenient and efficient to construct counterfactual pairs from existing downstream datasets.
>
> **Q1: For a context-embedded PLM, how do you obtain a contextual representation of tokens?**
>
> We use BERT-base-uncased in our experiments. Thus, for the single-sentence classification task, we prepend the [CLS] token before the input sentence s and feed it into the BERT model to get the embedding of the [CLS] token as sentence representation. For sentence-pair classification task, e.g., SNLI task, we prepend [CLS] before the premise x and [SEP] token to separate premise x and hypothesis y and feed it into the BERT model to get the embedding of [CLS] token as sentence representation.
>
> We will add this detailed explanation to the Appendix in Section A (More Implementation Details).
>
> **Q2. Explain the motivation for using prompts and what benefits it brings to Co$^2$PT.**
>
> 1) The previous debias-then-finetune methods require expensive re-training of the original PLM and risk knowledge forgetting. By incorporating continuous prompts as prefix tokens in every layer of the PLM, we can save computational and memory resources while preserving the original pre-trained knowledge and language modeling ability (Li and Liang, 2021; Liu et al., 2022).
>
> 2) In our main results (Tables 2-4), we show that Co$^2$PT achieves better performance on bias mitigation on downstream tasks. In addition, in Section 6 and in Table 5, we show that our prompt-based methods can be easily integrated with existing upstream debiased models; this is more efficient than full fine-tuning methods. Furthermore, without touching the parameters of debiased models, Co$^2$PT can preserve the debiased ability of existing upstream models while avoiding re-acquiring or amplifying bias on downstream tasks.
>
> **Q3. The debiasing of models often affects the performance of the model because counterfactual pairs are different from the real corpus, how did the authors balance the debiasing of the model with the performance?**
>
> 1) To balance the debiasing of the model with the model performance on downstream tasks, we simultaneously optimize the prompt tuning loss L_{pt} on downstream tasks, which represents the model performance on downstream tasks and contrastive loss L_{cl} between the counterfactual pairs, which aims to pull the counterfactual pairs closer, shown in Equation (2) in Line 280.
> 2) By performing a hyperparameter search to adjust the value of the tunable coefficient hyperparameter, we aim to find the appropriate hyperparameter that can achieve good bias mitigation performance while maintaining good downstream task performance, which is detailed described in Section 8 (Impact of Hyperparameters).
>
> **References**
>
> [1] Pengfei Liu, Weizhe Yuan, Jinlan Fu, Zhengbao Jiang, Hiroaki Hayashi, and Graham Neubig. 2023. Pre-train, Prompt, and Predict: A Systematic Survey of Prompting Methods in Natural Language Processing. ACM Comput. Surv. 55, 9, Article 195 (September 2023), 35 pages. https://doi.org/10.1145/3560815
>
> [2] Xiao Liu, Kaixuan Ji, Yicheng Fu, Weng Tam, Zhengxiao Du, Zhilin Yang, and Jie Tang. 2022. P-Tuning: Prompt Tuning Can Be Comparable to Fine-tuning Across Scales and Tasks. In Proceedings of the 60th Annual Meeting of the Association for Computational Linguistics (Volume 2: Short Papers), pages 61–68, Dublin, Ireland. Association for Computational Linguistics.

---

### Official Review · Reviewer_eamV · 2023-08-02

**Soundness:** 4

**Excitement:**

3: Ambivalent: It has merits (e.g., it reports state-of-the-art results, the idea is nice), but there are key weaknesses (e.g., it describes incremental work), and it can significantly benefit from another round of revision. However, I won't object to accepting it if my co-reviewers champion it.

**Paper Topic And Main Contributions:**

The paper proposes a method for debiasing pre-trained language models using prompt-tuning approach that relies on counterfactuals generated and augmented in the training data accompanying a contrastive loss in addition to objective for the downstream task. Unlike most previous works that debias the representations learned during pre-training, the authors mitigate the biases directly for the downstream task as they point out referring to previous work that biases might re-surface once model is fine-tuned. The method is evaluated on a sentence similarity task, natural language inference (NLI) task, and on Bias-In-Bios dataset involving occupation classification. In all these three datasets, the proposed method outperforms the baselines, often with a significant margin. The authors further show that the method can be augmented with the pre-trained debiased models to prevent the resurgence of biases during fine-tuning. Finally, authors present an ablation study for the components involved in their method and show both the counterfactual data as well as contrastive learning objective aid significantly to the performance of their method.

**Questions For The Authors:**

Line 378 (Equation 5), should there be a superscript TPR on GAP?

In Table 4, I am not sure how GAP^{TPR} is different from GAP^{RMS}. As defined in Equation 4,  GAP^{TPR}_{g, o} is calculated for each occupation and gender seperately. Is GAP^{TPR} simply obtained by averaging across all occupations and genders? If yes, does the only difference between GAP^{TPR} and GAP^{RMS} is that the absolute errors are averaged instead of the squared errors (followed by a square-root)? If yes, then the observed trends are slightly puzzling, as according to GAP^{TPR}  there is a reduction in bias over the BERT model directly fine-tuned, but that doesn't hold according to GAP^{RMS}.

**Reasons To Accept:**

1. The proposed method is intuitive, easy-to-use, and scalable. The results indicate that the method does a reasonable job in mitigating the downstream task biases while maintaining the task performance.
2. The experimental setup is quite detailed, with multiple datasets and baselines, as well as ablation studies and impact of different hyperparameters that help answer most of the questions the reader would have about the effectiveness of the proposed method. The claims of the paper are well supported with the experimental evidence. I have a few reservations about the choice of baselines which I detail more in the next question.
3. The paper has nice presentation, is written well and easy to follow along.

**Reasons To Reject:**

1. Baselines: To my best understanding all the baselines except perhaps ADELE are upstream debiased i.e. biases are mitigated in the pre-training representations and then those models are directly fine-tuned on the downstream tasks without any bias mitigation. To me this comparison sounds a bit unfair as the proposed method is fine-tuned to mitigate biases directly on the downstream task in contrast to the baselines. One obvious baseline that I can think of is performing CDA on the training data of downstream task and fine-tuning on the augmented data as in Zhao et al. 2018 [1] (a version of this is done in the ablation section when comparing with PT+CDA, but that versions performs CDA with prompt-tuning instead of full-fine tuning). In case, I have missed something here, I would be willing to revisit my stance on this point in the response period. Additionally, I feel the ZariCDA and ZariDO do not provide fair comparison as the base pre-trained model for these baselines is bert-large-uncased while for all other baselines and the proposed method it is bert-base-uncased. CDA and Dropout on bert-base-uncased instead would have been a more appropriate comparison. I can understand pre-training these models from scratch for these baselines can be quite expensive, but as it stands the comparison doesn't seem technically sound.

2. Mitigating biases beyond gender dimension: One of the advantages of the proposed method as highlighted by the authors is that it can be easily extended to different bias dimensions (gender, race, religion), without having to fine-tune the entire model. In that light, it would have been valuable to see the effectiveness of the method on the dimensions other than gender. I can understand, downstream tasks for bias evaluation might be sparse for other dimensions, but perhaps one can synthetically generate data (like in case of Bias-STS-B and Bias-NLI) for dimensions like race, and religion as well. The paper also claims that the proposed method can be effective to mitigate intersectional biases along these dimensions in downstream tasks, but again there are no experiments to support the claim.


[1] Jieyu Zhao, Tianlu Wang, Mark Yatskar, Vicente Ordonez, and Kai-Wei Chang. 2018. Gender Bias in Coreference Resolution: Evaluation and Debiasing Methods. In Proceedings of the 2018 Conference of the North American Chapter of the Association for Computational Linguistics: Human Language Technologies, Volume 2 (Short Papers), pages 15–20, New Orleans, Louisiana. Association for Computational Linguistics.

**Reproducibility:**

4: Could mostly reproduce the results, but there may be some variation because of sample variance or minor variations in their interpretation of the protocol or method.

**Reviewer Confidence:**

4: Quite sure. I tried to check the important points carefully. It's unlikely, though conceivable, that I missed something that should affect my ratings.

---

> ### Author Rebuttal · Authors · 2023-08-29
>
> We appreciate your positive comments that our proposed method is scalable and the paper is well-written with detailed experiments. In the following, we address your points in turn:
>
> **Weakness 1a) To my best understanding all the baselines except perhaps ADELE are upstream debiased models.**
>
> 1) As stated in Section 1 (Introduction), much prior effort has focused primarily on debiasing the representations learned during the pre-training process. Previous work is based on the hypothesis that if an upstream model is unbiased, it will also preserve its fairness effects on downstream tasks during the fine-tuning process. However, after fine-tuning the upstream debiased models on downstream tasks, our experimental results show that even if the upstream models have been debiased, those debiased models tend to re-acquire or even amplify biases during the fine-tuning process on downstream tasks, which is consistent with recent research findings and leads to our motivation – how to mitigate bias directly on downstream tasks more efficiently and effectively.
>
> 2) For baselines that are ***not upstream*** –  ADELE, PT+CDA, and FT+CDA (the one added in the next question), we notice that Co$^{2}$PT also shows better performance on bias mitigation on downstream tasks, demonstrating the effectiveness of our proposed methods.
>
> **Weakness 1b)  PT+CDA performs CDA with prompt-tuning instead of full-fine tuning.**
>
> 1) Co$^{2}$PT is a prompt-based method. Thus, we perform CDA with prompt tuning as an ablation study. This ablation study aims to evaluate the effectiveness and necessity of integrating the CDA module within the proposed Co$^{2}$PT framework.
>
> 2) Co$^{2}$PT applies prompt tuning instead of full fine-tuning on downstream tasks, thus it is ***not fair*** to compare it with full fine-tuning. In response to the reviewer's suggestions, we explore FT+CDA, which performs CDA with full fine-tuning BERT model on downstream tasks:
>
> |Model |Diff. $\downarrow$ | $\tau$:0.1 $\downarrow$ | $\tau$:0.3 $\downarrow$ | Pear./Spear. |
> |---|---|---|---|---|
> |FT | 0.282 | 0.867 | 0.417 | 0.883 / 0.879 |
> |**FT+CDA** | 0.131 | 0.511 | 0.080 | 0.885 / 0.881 |
> |Co$^{2}$PT | 0.058 | 0.167 | 0.005 | 0.884 / 0.880 |
>
> FT+CDA shows better performance on bias mitigation compared with the FT model (vanilla fine-tuning). These results underscore the effectiveness of CDA, mirroring the same tendency observed in PT and PT+CDA. Thus, incorporating these findings might appear *redundant*.
>
> We can still add the results of FT+CDA, update Table 6 (Impact of different components), and add analysis in Section 7 (Impact of Design). We will also add the standard deviation 0.020, 0.058, 0.042, 0.001 / 0.000 of FT+CDA to Table 11 in the appendix.
>
> **Weakness 1c) ZariCDA and ZariDO are bert-large-uncased instead of bert-base-uncased.**
>
> We kept the results of bert-large-uncased checkpoints for 3 reasons:
>
> 1) The authors of [4] did not release bert-base-uncased checkpoints, nor code to obtain those checkpoints (https://github.com/google-research-datasets/Zari);
> 2) [3] performed the same comparisons in their work (compare ADELE with ZariCDA and ZariDO);
> 3) For the baseline that finetunes BERT on downstream task STS-B, the BERT-large-uncased model achieves better task performance and less bias on Bias-STS-B.
>
> |Model |Diff. $\downarrow$ | $\tau$:0.1 $\downarrow$ | $\tau$:0.3 $\downarrow$ | Pear./Spear. |
> |---|---|---|---|---|
> BERT-base | 0.282 | 0.867 | 0.417 | 0.883 / 0.879 |
> **BERT-large** | 0.210 | 0.795 | 0.216 | 0.885 / 0.883
>
> Thus, the inclusion of these two baselines where the backbone models are BERT-large is actually ***advantageous***, as we learn that: while Co$^{2}$PT only uses BERT-base as the backbone model, it can achieve even better performance on bias mitigation than the larger model.
>
> **Weakness 2a: I can understand, that downstream tasks for bias evaluation might be sparse for other dimensions, but perhaps one can synthetically generate data (like in case of Bias-STS-B and Bias-NLI) for dimensions like race, and religion as well.**
>
> 1) The majority of previous works focus on one dimension (gender) [1,2,3,4,5]. We follow these previous works. We are interested in experimenting with our method in other dimensions (such as race and religion, as you suggest), and will update Section 9 to clarify our interest in this future work.
>
> 2) Just like you said, the sentences containing predefined word lists of race and religion (six pairs for each dimension in [6]) are really sparse in the STS-B and NLI datasets. Therefore, formatting Bias-STS-B and Bias-NLI in this same manner may not yield a good selection of data. We will need to experiment with other, additional datasets in the future to investigate our method in these other dimensions (race, religion, etc.).
>
> **Weakness 2b: There are no experiments to support the claim to mitigate intersectional biases along these dimensions in downstream tasks.**
>
> To make it clearer, this is the discussion of our future work, which we put in the *"Conclusion and Discussion"* section. Based on the results we get in the paper, the structure of our framework is flexible and has the potential to be applied to different dimensions, e.g., religion, race, and intersectional biases. Specifically, we can apply Co$^{2}$PT to train different continuous prompts for gender, religion, and race separately, and then concatenate the prompts to determine whether or not it can mitigate the intersectional biases on downstream tasks. On the opposite, previous methods need to train different individual models for different bias dimensions, which is time-consuming and expensive. We hypothesize that the generic structure of Co$^{2}$PT may make it more suitable and efficient for mitigating intersectional biases.
>
> We will update the title of Section 9 to *"Conclusion and Future Work Discussion"*.
>
> **Question 1(a): Should there be a superscript TPR on GAP?**
>
> Yes, thank you for asking about this. We will update it.
>
> **Question 1(b): I am not sure how GAP$^{TPR}$ is different from GAP$^{RMS}$.**
>
> Following previous work [1,2] GAP$^{TPR}$ denotes the gap between the true positive rates of the male prediction results and the female prediction results over all occupations, while the GAP$^{RMS}$ represents the root mean square of the true positive rates difference for each occupation.
>
> The motivation for using the root mean square follows from previous work: larger values have a larger effect in GAP$^{RMS}$. We will make it more clear and concise by updating the descriptions and subscripts in equation (4) from GAP_{g,o}^{TPR} = |TPR_{g,o} - TPR_{\~g,o}|
>   to GAP^{TPR} = |TPR_{g} - TPR_{\~g}|.
>
> **References**
>
> [1] Jacqueline He, Mengzhou Xia, Christiane Fellbaum, and Danqi Chen. 2022. MABEL: Attenuating Gender Bias using Textual Entailment Data. In Proceedings of the 2022 Conference on Empirical Methods in Natural Language Processing, pages 9681–9702, Abu Dhabi, United Arab Emirates. Association for Computational Linguistics.
>
> [2] Yingji Li, Mengnan Du, Xin Wang, and Ying Wang. Prompt Tuning Pushes Farther, Contrastive Learning Pulls Closer: A Two-Stage Approach to Mitigate Social Biases. ACL 2023.
>
> [3] Anne Lauscher, Tobias Lueken, and Goran Glavaš. 2021. Sustainable Modular Debiasing of Language Models. In Findings of the Association for Computational Linguistics: EMNLP 2021, pages 4782–4797, Punta Cana, Dominican Republic. Association for Computational Linguistics.
>
> [4] Kellie Webster, Xuezhi Wang, Ian Tenney, Alex Beutel, Emily Pitler, Ellie Pavlick, Jilin Chen, Ed Chi, and Slav Petrov. 2020. Measuring and reducing gendered correlations in pre-trained models. arXiv preprint arXiv:2010.06032.
>
> [5] Masahiro Kaneko and Danushka Bollegala. 2021. Debiasing Pre-trained Contextualised Embeddings. In Proceedings of the 16th Conference of the European Chapter of the Association for Computational Linguistics: Main Volume, pages 1256–1266, Online. Association for Computational Linguistics.
>
> [6] Nicholas Meade, Elinor Poole-Dayan, and Siva Reddy. 2022. An Empirical Survey of the Effectiveness of Debiasing Techniques for Pre-trained Language Models. In Proceedings of the 60th Annual Meeting of the Association for Computational Linguistics (Volume 1: Long Papers), pages 1878–1898, Dublin, Ireland. Association for Computational Linguistics.

---

### Meta-Review · Area_Chair_Qr59 · 2023-09-19

**Recommendation:** 3

**Metareview:**

The paper "Co^2PT: Mitigating Bias in Pre-trained Language Models through Counterfactual Contrastive Prompt Tuning" presents work on debiasing through prompting  and testing this method based on three down-stream tasks.

Points mentioned for rejecting the paper refer to the experimental setup.
Arguments for the paper is the clearly written and detailed information given, but also the approach taken.

---

### Decision · Program_Chairs · 2023-10-07

**Decision:**

Accept-Findings

**Comment:**

The paper "Co^2PT: Mitigating Bias in Pre-trained Language Models through Counterfactual Contrastive Prompt Tuning" presents work on debiasing through prompting  and testing this method based on three down-stream tasks.

Points mentioned for rejecting the paper refer to the experimental setup.
Arguments for the paper is the clearly written and detailed information given, but also the approach taken.